# Effects of Different Intensity Exercise on Glucose Metabolism and Hepatic IRS/PI3K/AKT Pathway in SD Rats Exposed with TCDD

**DOI:** 10.3390/ijerph182413141

**Published:** 2021-12-13

**Authors:** Huohuo Wang, Juanjuan Wang, Yihua Zhu, Huiping Yan, Yifan Lu

**Affiliations:** 1School of Sports Medicine and Rehabilitation, Beijing Sport University, Beijing 100084, China; wanghuohuo123@outlook.com (H.W.); xjwang2009@126.com (J.W.); zhuyihua2@163.com (Y.Z.); 2Key Laboratory of Sports and Physical Fitness of the Ministry of Education, Beijing Sport University, Beijing 100084, China

**Keywords:** TCDD, exercise, insulin, IRS/PI3K/AKT pathway

## Abstract

The objective of the study was to investigate the effects of different intensity exercise and 2,3,7,8-Tetrachlorodibenzo-p-dioxin (TCDD) exposure on glucose metabolism in Sprague Dawley (SD) rats, as well as the action of insulin receptor substrate (IRS)/phosphatidylinositol-3-kinases (PI3K)/protein kinase (AKT) signaling pathway in it. Besides that, we explored whether exercise can alleviate the toxicity induced by TCDD. Sixty male SD rats (8 weeks old) were randomly divided into non-exercise group, none-exercise toxic group, moderate-intensity exercise group, moderate-intensity exercise toxic group, high-intensity exercise group, high-intensity exercise toxic group. The toxic groups were intraperitoneally injected with TCDD, which the dose was 6.4 µg/kg· BW for the first week, then 21% of the above week dose for continuous 8 weeks. The 8-week treadmill running of moderate intensity (15 m/min, 60 min/day) and high intensity (26 m/min, 35 min/day) were implemented separately in exercise groups five times a week. After detecting the concentration of fasting serum glucose, insulin and C-peptide, the index of the homeostasis model assessment of insulin resistance (HOMA-IR) and islet β-cell secretion (HOMA-β) were calculated. We measured the hepatic mRNA expression levels of IRS2, phosphatidylinositol-3-kinases catalytic subunit alpha (PIK3CA), AKT by real-time PCR. The protein expression of total IRS2 (tIRS2), phosphorylated IRS2 at Ser731 (pSer731), total PIK3CA (tPIK3CA), total Akt (tAkt), phosphorylated Akt at Thr308 (pThr308) in liver were analyzed by western blot. We observed that compared to the non-exercise group, insulin and HOMA-IR index were significantly higher in the none-exercise toxic group (*p* < 0.05), while glucose, insulin, C-peptide and HOMA-IR index were significantly lower in the moderate-intensity exercise group (*p* < 0.05). In the high-intensity exercise group, the HOMA-IR index was significantly lower and the gene expression of IRS2 was significantly higher than in the non-exercise group (*p* < 0.05). Besides that, the HOMA-β index in the moderate-intensity exercise toxic group was significantly higher compared to the none-exercise toxic group and moderate-intensity exercise group (*p* < 0.05). The level of IRS2mRNA was significantly lower in the high-intensity exercise toxic group than in the high-intensity exercise group (*p* < 0.05). Our results demonstrated that 8-week TCDD exposure could induce insulin resistance in rats, while exercise could improve insulin sensitivity in which moderate intensity was more obvious than high intensity exercise. Meanwhile, both intensity exercise could not effectively alleviate the insulin resistance induced by TCDD, but high intensity exercise could promote compensatory insulin secretion to maintain glucose homeostasis. Although the gene expression of IRS2 was changed in high-intensity exercise groups, the mediation role of the hepatic IRS2/PI3K/AKT pathway in the effects of exercise and TCDD exposure on glucose metabolism remains very limited.

## 1. Introduction

One of the persistent organic pollutants (POPs) which is widely used in daily life and causes a variety of toxic effects is 2,3,7,8-Tetrachlorodibenzo-p-dioxin (TCDD). TCDD mainly collects in liver and adipose tissue after entering the human body. Previous epidemiological studies provide sufficient evidence for a positive association between POPs (including TCDD) and diabetes mellitus type 2 (T2DM) [1,2]. A correlation study demonstrated that exposure to TCDD is associated with hyperinsulinemia and insulin resistance [3]. Several experimental studies have provided some important insights that by binding to the aryl hydrocarbon receptor (AhR), TCDD causes changes in translational and transcriptional mechanisms resulting in glucose transporter (GLUT) expression decreased and insulin resistance [4,5,6]. However, the detailed molecular mechanism underlying TCDD-induced insulin resistance is yet to be elucidated.

Insulin resistance, a condition in which responders are insensitive to insulin, is mainly found in insulin-sensitive tissues such as liver, muscle and adipose tissue [7]. Impaired insulin signaling is the direct reason for insulin resistance. The most classic insulin signaling pathway is the insulin receptor substrate (IRS)/phosphatidylinositol-3-kinases (PI3K)/protein kinase (AKT) regulatory pathway. The pathway is activated when insulin binds to the insulin receptor (InsR) on the membrane. Activated InsR is able to lead to tyrosine phosphorylation of IRS and its related substrate protein family, mainly IRS-1/2 [8]. IRS can activate PI3K to produce second messengers to stimulate downstream of Akt [9,10,11]. Insulin-stimulated Akt results in the uptake of circulating glucose through GLUT4 translocation from intracellular compartments to the cell membrane [12]. Therefore, insulin signaling is accomplished through the activation process of a series of proteins and molecules such as InsR, IRS-1/2, PI3K, Akt and GLUT4. Alterations in any of them throughout the pathway can trigger insulin resistance.

According to the report of the World Health Organization (WHO) on diabetes, regular exercise is an effective avenue to prevent diabetes and delay the course of it [13]. A large amount of research provides evidence that structured exercise is an effective interventional strategy to improve glucose control and alleviate insulin resistance in T2DM [14]. However, the effectiveness of exercise depends on its intensity. Mikael etal. has reported that excessive exercise training causes mitochondrial functional impairment and decreases glucose tolerance in healthy volunteers [15]. Therefore, it is very important to choose proper intensity exercise to improve insulin sensitivity. Our research groups have previously reported that exercise can effectively relieve the toxicity of lipid metabolism and oxidative stress in the liver of Sprague Dawley (SD) rats exposed with TCDD [16,17,18]. Since lipid metabolism disorder and oxidative stress both contribute to the result of insulin resistance [19,20], these observations could be of particular interest in the case of glucose metabolism and underlying IRS/PI3K/Akt signaling pathway.

In view of the above consideration, the aim of the present research was to investigate the effects of different intensity exercise or TCDD exposure on glucose metabolism, as well as the role of IRS/PI3K/Akt signaling pathway in that. In addition, whether exercise can alleviate the toxicity of glucose metabolism and pathway induced by TCDD is also an important concern. The results of this study will provide new insights into preventing diabetes effectively by proper exercise under present environmental pollution conditions.

## 2. Materials and Methods

### 2.1. Subjects

Sixty male SD rats aged 7 weeks were housed in standard cages under controlled conditions (cleaning grade of SPF, 40–70% humidity, 20–26 ℃, and an alternating 12 h light–dark cycle). They were fed with the conventional rat chow of national standard, free drinking water and diet. This experiment was approved by the Animal Ethics Committee of Beijing Huafukang Biotechnology Co., Ltd, Beijing, China (IACUC-20190602). 

### 2.2. Experimental Protocol

After acclimatizing the 60 rats to their environment for one week, they were divided into 6 groups according to their body weight which was all normal, with no difference among groups, as follows: non-exercise group (NC), none-exercise toxic group (NT), moderate-intensity exercise group (MIT), moderate-intensity exercise toxic group (MIT+T), high-intensity exercise group (HIT), high-intensity exercise toxic group (HIT+T). They were imposed TCDD exposure protocol or/and exercise protocol in continuous 8 weeks (Figure 1).

The 2,3,7,8-TCDD standard solution (98% purity), purchased from Cambridge Isotope Laboratories (Andover, MA, USA) was dissolved in corn oil. It was intraperitoneally injected into rats of NT, MIT+T and HIT+T groups with the exposure dose of 6.4 µg/kg· BW for the first week, then 21% of the above week dose in continuous 8 weeks [21]. The corresponding control rats in NC, MIT and HIT groups were injected with corn oil in the same way.

Two different intensity treadmill running interventions were conducted in four exercise groups five days per week for 8 weeks, while rats in the NC group and NT group kept rest. According to the classical model of treadmill exercise established by Bedford et al. on 10-week-old SD rats [22], the program of the exercise interventions was formulated in Table 1.

### 2.3. Serum and Liver Collection

At week 8, the rats were sacrificed after 48 h from the last invention of the exercise groups. Before that they were deprived of food and water for one day. After being weighed, rats were anesthetized by intraperitoneal injection of 7% chloral hydrate at the dose of 0.5 mL/100 g·bw. The blood was collected with negative pressure tubes and centrifuged at 3000 rpm for 15 min to obtain serum. The liver was dissected, and the segments of liver were flash frozen in liquid nitrogen immediately. All the samples were divided in order and stored in a refrigerator at −80 ℃.

### 2.4. Determination of Serum Glucose, Insulin and C-Peptide

Glucose oxidase method was used to measure the fasting serum glucose concentration of rats (kits from Zhongshengbeikong Biotechnology, 180851, Beijing, China). The concentrations of serum insulin and C-peptide protein were measured by enzyme-linked immunoassay (kits from Huamei Bioengineering, CSB-E05070r and CSB-E05067r, Wuhan, China). The indexes of the homeostasis model assessment of insulin resistance (HOMA-IR) and islet β-cell secretion (HOMA-β) were calculated according to the result of glucose and insulin [23].
HOMA-IR = FINS × FPG/22.5 
HOMA-β = 20 × FINS/(FPG − 3.5)

(Note: Fasting plasma glucose-FPG (mmol/L), Fasting plasma insulin-FINS(μU/mL)).

### 2.5. Realtime-PCR

Guided by the manufacturer’s instructions, hepatic cDNAs of IRS2/phosphatidylinositol-3-kinases catalytic subunit alpha (PIK3CA)/AKT were synthesized (Kits from TIANGEN, DP431, Beijing, China). Realtime PCR was performed with the Applied Biosystems (Thermo Scientific™, ABI7500, Waltham, MA, USA) using the SYBR^®^ Premix ExTaqTM II (Takara, RR820A, Dalian, China). The sequences of primers (synthesized by Shenggong Bioengineering, Shanghai, China) are shown in Table 2 and their effectiveness were verified by the BLAST function test on PubMed. The mRNA expression was normalized against the corresponding control gene β-actin. Relative quantitative method was used to calculate the expression levels of the three genes, and the relative multiple change of the target gene = 2^−^^ΔΔCT^: 2^−^^ΔΔCT^ = [(CT_target gene_ − CT_β-actin_) experimental group − (CT_target gene_ − CT_β-actin_) corresponding control group]

### 2.6. Western Blot

Individually-isolated total protein (40 μg or 80 μg) was separated on 10% SDS- polyacrylamide gels and transferred to Polyvinylidene Fluoride Membrane. The following primary rabbit monoclonal or polyclonal antibodies were used: total IRS2 (tIRS2) (diluted 1:1000, Abcam, ab134101, Cambridge, UK), phosphorylated IRS2 at Ser731 (pSer731) (diluted 1:5000, Abcam, ab3690, Cambridge, UK), total PIK3CA(tPIK3CA) (diluted 1:1000, Abcam, ab40776, Cambridge, UK), total Akt (tAkt) (diluted 1:500, Abcam, ab8805, Cambridge, UK), phosphorylated Akt at Thr308 (pThr308) (diluted 1:500, Abcam, ab38449, Cambridge, UK). Detection of signals was performed using the EasySee^®^ Western Blot Kit (TransGen, DW101, Beijing, China) with Goat Anti-Rabbit HRP Conjugated IgG (TransGen, HS101, Beijing, China) as the second antibody. Polyclonal rabbit β-actin antibody (TransGen, HC201, Beijing, China) was used as the loading control to normalize the signal obtained for proteins. The density of immunoreactive bands was measured using ImageJ 1.52. The corresponding control levels were arbitrarily assigned a value of 1.

### 2.7. Data Analysis

The digital statistics were expressed as mean± standard deviation (Mean ± SD). All statistical analyses were conducted by SPSS (Version 25). The data of body weight were analyzed by repeated measurement ANOVA and other data were analyzed by two-factor ANOVA. Significant difference was considered as *p* < 0.05, and *p* < 0.01 was considered as very significant difference.

## 3. Results

### 3.1. Body Weight in Rats

In the 1–7 weeks, the body weight of experimental rats in all groups continued to increase overtime, with significant differences between any two weeks in the same group (*p* < 0.001) (Figure 2). There was an interaction between TCDD exposure and time (*p* < 0.001). The weight of rats in the NT group stopped increasing in the last week, and there was no significant difference between the last two weeks. There was also an interaction between exercise and time (*p* < 0.001). After exercise with moderate or high intensity, the body weight of the rats increased for 7 weeks but decreased at 8th week, leading to the weight approaching to that in 6th week. There was no interaction between exercise and TCDD exposure. Two-factor ANOVA was conducted to compare the weight of rats in different groups at 8th week, and it was found that the body weight in exercise groups (*p* < 0.001) and TCDD exposure groups (*p* < 0.001) are significantly lower than in the NC group. The weight decrease of rats in high intensity exercise groups was more obvious than that of moderate intensity exercise groups.

### 3.2. Serum Biochemical Parameters in Rats

In the data of serum biochemical parameters in rats (Figure 3), fasting serum glucose (*p* = 0.022), insulin (*p* = 0.041) and C-peptide (*p* = 0.011) were all significantly lower in the MIT group while insulin concentration was very significantly higher in the NT group (*p* = 0.002) compared to the NC group. The index of HOMA-IR was significantly lower in the MIT group (*p* = 0.001) and HIT group (*p* = 0.021) compared to the NC group, and the change was more obvious in the MIT group than in the HIT group. However, the HOMA-IR index was very significantly higher in the NT group than in the NC group (*p* = 0.003). Exercise and TCDD exposure have an interaction relationship on the HOMA-β index. Compared to the MIT group, the HOMA-β index of the MIT+T group was very significantly higher (*p* = 0.004). Similarly, compared to the NT group, the HOMA-β index of the MIT+T group was significantly higher (*p* = 0.049).

### 3.3. Hepatic Gene/Protein Expression Profile in Rats

We next examined the levels of various genes expression in different groups by real-time PCR in liver (Figure 4). In the data of IRS2mRNA, there was an interaction relationship between exercise and TCDD exposure (*p* = 0.036), and the significance was mainly in the high-intensity exercise groups. Compared to the NC group, the level of IRS2mRNA in the HIT group was significantly higher (*p* = 0.049). Meanwhile in the HIT+T group, the expression of IRS-2 mRNA was very significantly lower than in the HIT group (*p* = 0.001). Other data showed no significant change.

There is not any significant change in the expression or phosphorylation of proteins in TCDD exposure or/and exercise groups compared to the NC group (Figure 5).

## 4. Discussion

### 4.1. Glucose Metabolism

Fasting glucose and insulin value are very intuitive markers reflecting the status of glucose metabolism in the body, and their level can indicate insulin resistance and the function of pancreatic β-cells to some extent. In the course of T2DM, insulin usually rises in the early stages and falls in the later stages [24]. Meanwhile, C-peptide is released in the same way with insulin but more stable, thus the function of insulin secretion of pancreatic β-cells can be assessed more precisely by it [25]. In addition, the index of HOMA-IR and HOMA-β proposed by Matthews et al. are widely used to measure insulin resistance and insulin secretory capacity with a strong validity in clinical and experimental studies [23].

In the present study, the final body weight of rats performing exercise was lighter. This may be due to the energy expenditure effect of exercise. The rats performing high-intensity exercise had lighter body weight than those with moderate-intensity exercise, which also indicated that more vigorous exercise is more effective in controlling body weight. As for the glucose metabolism, exercise can decrease the level of glucose, insulin, C-peptide and HOMA-IR. However, these effects mainly depended on moderate intensity exercise, and high intensity exercise only imposed significant change in HOMA-IR. It demonstrates that exercise can improve insulin sensitivity, and the effect of moderate intensity exercise is more obvious than that of high intensity exercise. 

Studies on whether there is difference in the effect of different intensity exercise on insulin sensitivity have been inconclusive and remain controversial. Emily et al. reported that two weeks of exercise training at either moderate or high intensity resulted in decreasing the level of insulin resistance and increasing insulin sensitivity in older, overweight individuals with prediabetes, with no difference between two exercise intensities [26]. In the present study, although the effect of moderate-intensity exercise was more obvious than that of high-intensity exercise on insulin sensitivity, there was no significant difference between themselves. The different results of studies may be related to the duration of training, the time of measurement after the last training session, and the method of measuring insulin sensitivity [27].

In the results of the study, TCDD exposure resulted in significant increase in insulin and HOMA-IR index indicating the occurrence of insulin resistance or insulin sensitivity decreased in rats. Besides that, we observed that TCDD exposure slowed down the speed of weight gain of rats, leading to the final body weight being lighter than that of rats in the NC group. Considering the results above, TCDD exposure may have caused metabolic syndrome in rats, which is the most prominent symptom of TCDD-induced toxicity [28].

Notably, TCDD and exercise imposed opposite effects on insulin resistance or insulin sensitivity in rats, but there was no significant interaction between them, indicating that exercise could not alleviate insulin resistance induced by TCDD. However, there was an interesting phenomenon that high intensity exercise interacted with TCDD on the HOMA-β index, reflecting that the function of pancreatic β-cells was enhanced. In previous studies, HOMA-IR seems to reflect insulin resistance well, but the relationship between HOMA-β cells and the true function of pancreatic β cells remains open to discussion. For example, in patients without insulin resistance, lower levels of insulin secretion do not negatively affect health because they produce insulin to clear glucose more efficiently; in contrast, in some patients with insulin resistance, the same insulin does not clear blood glucose efficiently, and these individuals require large amounts of insulin for similar glucose clearance, which leads to higher levels of insulin secretion [29]. Therefore, the functional level of pancreatic β-cells is influenced by both their own secretory capacity and the status of insulin resistance.

In the present study, the HOMA-β index was significantly higher in the MIT+T group compared to that in the MIT group, which was mainly caused by insulin resistance induced by TCDD, resulting in abnormal insulin secretion. However, in the groups of NT and MIT+T, performing moderate intensity exercise also elevated the HOMA-β index. The reason for this may be that in healthy rats, exercise actives insulin signaling pathway and enhances insulin sensitivity, resulting in decreased insulin and pancreatic β-cell secretion; whereas in our experimental protocol, TCDD exposure preceded exercise intervention for one day, which possibly means that exercise began when the insulin transduction pathway has been severely impaired by TCDD. In this case, exercise cannot enhance insulin sensitivity or reverse insulin resistance. Considering that exercise can promote the secretory function of islet β-cells themselves [30], we conclude that insulin secretion was enhanced for compensation to maintain glucose homeostasis after exercise. There is much controversy regarding the effect of exercise on insulin secretion while insulin resistance is occurring. Most of the findings show a decrease in insulin secretion level in obese people or people with insulin resistance after performing exercise [29], but Huang et al. found that exercise for three days after implementing the diabetes inducer STZ (Streptozotocin) injection tripled insulin level in rats in the later stages of the experiment [31]. Howarth et al. found no change in insulin level in diabetic rats after running exercise one week after STZ injection compared to the control group [32]. This suggests that exercise is protective on insulin pathways if it is started before the onset of disease; however, exercise after the onset of severe insulin resistance may be ineffective, only improving insulin secretion for compensation to maintain blood glucose homeostasis. Perhaps it is recommended that exercise before being severely disturbed by TCDD is more effective in preventing and mitigating the effects caused by the toxin.

### 4.2. The Action of the IRS/PI3K/Akt Signaling Pathway on Glucose Metabolism in Rats

Upon entry into the hepatic organism, TCDD promotes the expression of the cytochrome P450 enzyme via AhR, which is most typically characterized by inducing the toxicity of oxidative stress [33]. Oxidative stress can lead to the activation of several inflammatory pathways, such as nuclear factor kappa-B (NFκB). The activation of serine/threonine kinases required for NFκB activation can also lead to serine/threonine phosphorylation of IRS-1/2 without normal tyrosine phosphorylation, which disrupting the downstream protein cascades and leading to blocked insulin signaling [34]. The insulin receptor substrate family of proteins play a central role in insulin signaling. Unlike the expression of IRS1 in muscle and adipose tissue, IRS2 is a major player of insulin action in the liver [35]. Phosphorylation at Ser731 is counter regulatory, thus it can be used as a marker to evaluate the impaired insulin signaling mediated by IRS2 disruption [36].

In the result of gene expression of our study, only the statistics of IRS2 were significantly changed, while PI3K and AKT were not. We observed that high-intensity exercise elevated IRS2 gene expression, but combining with TCDD exposure significantly decreased its expression. This suggests that separate high-intensity exercise can positively promote IRS2 gene expression, but TCDD may have a suppressive effect. Their effects failed to induce any change of mRNAs of PI3K and AKT. As for the results of the protein and phosphorylation expression, TCDD exposure and exercise both failed to impose any effect on them.

The result demonstrated that the target of the positive effect of exercise on insulin pathway is the gene expression of IRS2, and the effect is only relied on high intensity exercise. Oxidative stress induced by high-intensity exercise may be a potential cause for this result. Activating the cascade with oxidative stress increased IRS2 gene promoter activity substantially [37]. In our research, exercise did not exert any effect on the protein expression and phosphorylation of IRS/PI3K/AKT, yet previous studies related this are not consistent. A cross-sectional study comparing trained and sedentary subjects found that exercise training was associated with an increase in muscle tissue on protein expression of IRS-1 and IRS-2, with no effect on Akt [38]. The reasons for this may be related to the fact that the effect of exercise on the insulin pathway acts directly on skeletal muscle rather than liver. In addition to that, large amount of researches have reported that exercise would mediate insulin signaling pathways throughout the activation of Adenosine 5‘-monophosphate (AMP)-activated protein kinase (AMPK)/acetyl-CoA carboxylase (ACC) [39]. These reasons may have contributed to the negative results of exercise on the protein expression and phosphorylation of IRS/PI3K/AKT in this study. As for the effect of TCDD on the insulin pathways, only when combined with high intensity exercise did TCDD have a suppressive effect. This can be explained by the reason that when performing no exercise or moderate intensity exercise, oxidative stress induced by TCDD exposure may be not enough to cause changes in IRS/PI3K/AKT pathways. By contrast, considering that high intensity exercise could induce responsive oxidative stress by itself, the oxidative stress as well as inflammatory action of TCDD exposure may be magnified by it leading to the decrease in the gene expression of IRS2. However the effect is not sufficient to induce any change in the downstream of the pathway and glucose metabolism.

## 5. Conclusions

Continuous TCDD exposure for eight weeks could induce insulin resistance and glucose metabolism disorder in rats. Moderate intensity exercise could improve insulin sensitivity in rats more obviously than high intensity exercise. Meanwhile, both moderate and high intensity exercise could not effectively alleviate the insulin resistance caused by TCDD, but high intensity exercise could promote compensatory insulin secretion to maintain glucose homeostasis. Although the expression of IRS2 gene was significantly changed by high-intensity exercise or combined with TCDD, the mediation role of the hepatic IRS2/PI3K/AKT pathway in the effect of exercise and TCDD exposure on glucose metabolism remains very limited, and other pathways related to insulin action need to be further elucidated.

## Figures and Tables

**Figure 1 ijerph-18-13141-f001:**
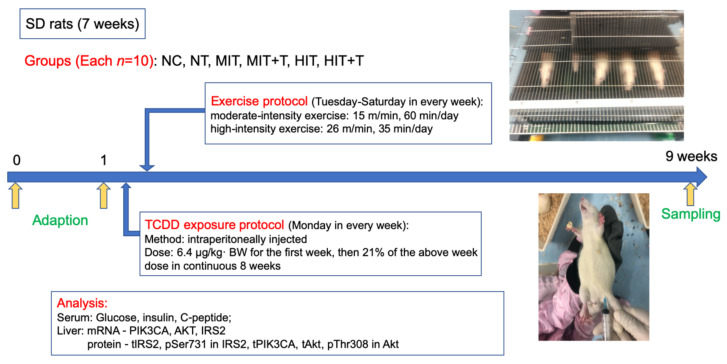
Schematic of experimental design. NC, non-exercise group; NT, none-exercise toxic group; MIT, moderate-intensity exercise group; MIT+T, moderate-intensity exercise toxic group; HIT, high-intensity exercise group; HIT+T, high-intensity exercise toxic group.

**Figure 2 ijerph-18-13141-f002:**
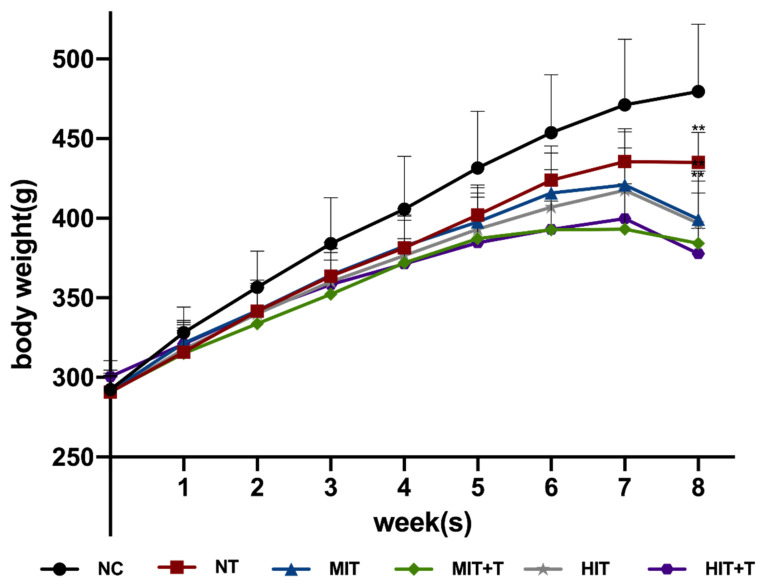
Changes in body weight of rats in continuous 8 weeks. **, *p* < 0.01 vs. the NC group in 8th week. NC, non-exercise group; NT, none-exercise toxic group; MIT, moderate-intensity exercise group; MIT+T, moderate-intensity exercise toxic group; HIT, high-intensity exercise group; HIT+T, high-intensity exercise toxic group.

**Figure 3 ijerph-18-13141-f003:**
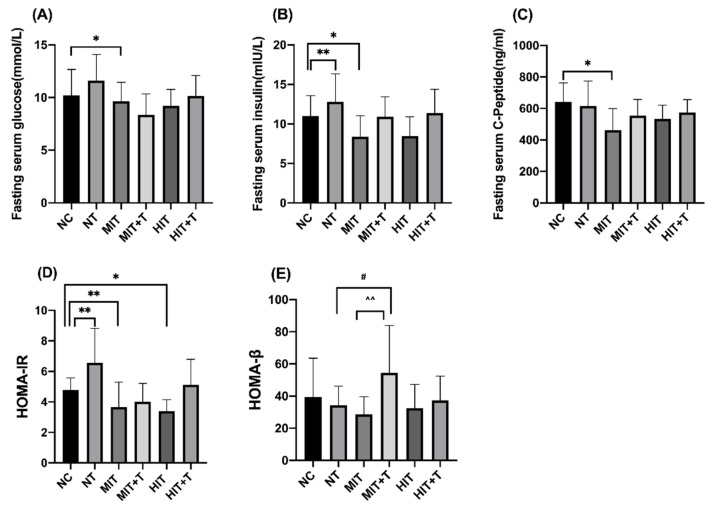
Changes of fasting serum glucose (**A**), insulin (**B**), C-peptide (**C**), the index of the homeostasis model assessment of insulin resistance (HOMA-IR, **D**) and islet β-cell secretion (HOMA-β, **E**) of six groups. Statistics are means ± SD. *, *p* < 0.05 vs. the NC group; **, *p* < 0.01 vs. the NC group; #, *p* < 0.05 vs. the NT group; ^^, *p* < 0.01 vs. the MIT group. NC, non-exercise group; NT, none-exercise toxic group; MIT: moderate-intensity exercise group; MIT+T, moderate-intensity exercise toxic group; HIT, high-intensity exercise group; HIT+T, high-intensity exercise toxic group.

**Figure 4 ijerph-18-13141-f004:**
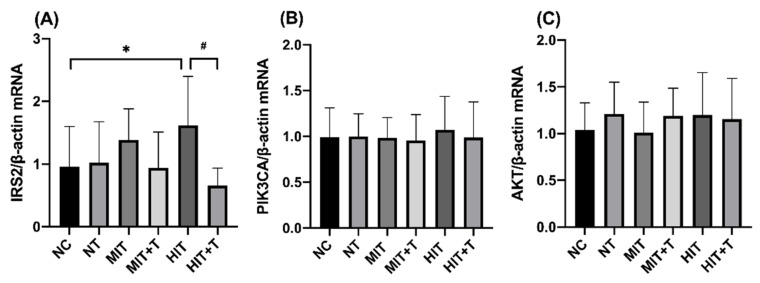
Hepatic expression of mRNAs by real-time PCR of insulin receptor substrate 2(IRS2, **A**), phosphatidylinositol-3-kinases catalytic subunit alpha (PIK3CA, **B**), protein kinase (AKT, **C**). Results represented the means ± SD. *, *p* < 0.05 vs. the NC group; #, *p* < 0.05 vs. the HIT group. NC, non-exercise group; NT, none-exercise toxic group; MIT: moderate-intensity exercise group; MIT+T, moderate-intensity exercise toxic group; HIT, high-intensity exercise group; HIT+T, high-intensity exercise toxic group.

**Figure 5 ijerph-18-13141-f005:**
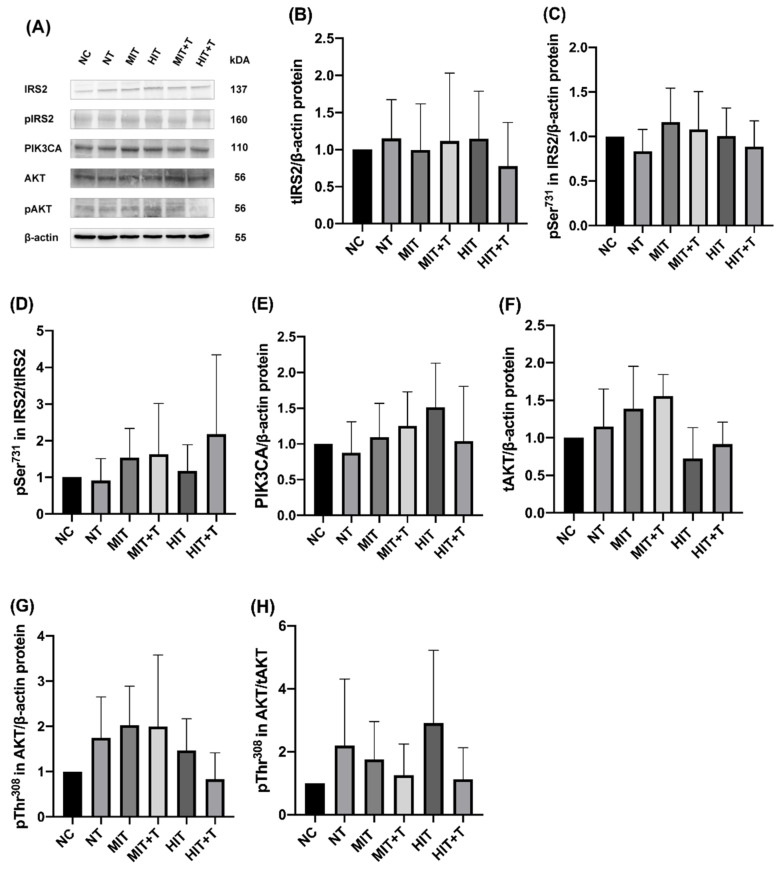
Western blot bands of total IRS2(tIRS2), phosphorylated Ser731 in IRS2 (pSer731 in IRS2), total PIK3CA(tPIK3CA), total AKT(tAKT), phosphorylated Thr308 in AKT(pThr308 in AKT) (**A**). Hepatic protein expression level of tIRS2 (**B**) and pSer731 in IRS2 (**C**). The ratio of pSer731 in IRS2 to tIRS2 (**D**). The protein expression of tPIK3CA (**E**), tAKT (**F**), pThr308 in AKT (**G**). The ratio of pThr308 in AKT to tAKT (**H**). The corresponding control levels were arbitrarily assigned a value of 1. Data are means ± SD. NC, non-exercise group; NT, none-exercise toxic group; MIT, moderate-intensity exercise group; MIT+T, moderate-intensity exercise toxic group; HIT, high-intensity exercise group; HIT+T, high-intensity exercise toxic group.

**Table 1 ijerph-18-13141-t001:** Exercise protocol.

Groups	Grade	Speed (m/min)	Time (min/day)	Distance (m)	VO2
MIT	5%	15	60	900	60%VO2max
MIT+T	5%	15	60	900	60%VO2max
HIT	5%	26	35	910	75%VO2max
HIT+T	5%	26	35	910	75%VO2max

**Table 2 ijerph-18-13141-t002:** The primer sequences of IRS2, PIK3CA and Akt.

Gene	Primer Sequence		Product Size (bp)
	Forward	Reverse	
IRS2	CTGGACAGAGGACTGAGGAGAGG	AGGCAGAGGAAGGCTGAGGAAC	82
PIK3CA	AGGATGCCCAACTTGATGCTGATG	CCGTTCATATAGGGTGTCGCTGTG	119
AKT	CAGGAGGAGGAGAGATGGACTTC	CACACGGTGCTTGGGCTTGG	99
β-actin	CCCAGGCATTGCTGACAGGATG	TGCTGGAAGGTGGACAGTGAGG	144

## Data Availability

The data presented in this study are available on request from the corresponding author.

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
