# Peer review of "Effects of Different Intensity Exercise on Glucose Metabolism and Hepatic IRS/PI3K/AKT Pathway in SD Rats Exposed with TCDD"

_ijerph, 2021, doi:10.3390/ijerph182413141_

Round 1

Reviewer 1 Report

The manuscript of Wang et al,  turns out to be interesting, even if there are some points to be clarified.
In the abstract the authors should specify the acronyms, as well as in the keywords.
Why did they only choose male mice?
Couldn't TCDD have a greater effect on females?
In M&M specify whether the mice are of normal weight or not.
In the M&M specify the codes of the kits used.
In fig 2 the significance between the various groups is not very clear, please describe better.
Figure 3A does not correspond to the relative histograms.
Furthermore, a higher resolution WB figure should be inserted.
It would have been interesting to include another endocrine disruptor, what do the authors think about it?
The discussion is too dispersive and the pathway in relation to physical activity and glucose metabolism should be better clarified, in this regard the authors could refer to this work (Treatment with sera from Water Polo athletes activates AMPKα and ACC proteins In HepG2 hepatoma cell line). 

Author Response

Response to Reviewer 1 Comments

Point 1: 
In the abstract the authors should specify the acronyms, as well as in the keywords.

Response 1: All the acronyms in abstract have been provided the full name. But in keywords, the full name was too long to write properly, and I find it is not mandatory in other papers published in this journal. Thus I did not specify the acronyms in the part of keywords. Please pardon me.

Point 2: Why did they only choose male mice? Couldn't TCDD have a greater effect on females?

Response 2: Admittedly, there are studies having reported that TCDD has sex-dependent effects on metabolism and studies in males should not be used to predict the effects of pollutants in females (e.g.: Long-term metabolic consequences of acute dioxin exposure differ between male and female mice.). But the evidence is not sufficient and it still needs further verification. Most researches regarding to TCDD exposure and exercise are mainly on male rodent. In addition to that, the objective of our research is investigating the effects of different intensity exercise and TCDD exposure on glucose metabolism, as well as their interaction on that. Thus we chose the most classic male rodent model to make a preliminary exploration. We will consider the sex-specific metabolic outcomes in rodents in our future research.

Point 3: In M&M specify whether the mice are of normal weight or not.

Response 3: All the rat are of normal weight and this is added in line 118.

Point 4: In the M&M specify the codes of the kits used.

Response 4: All the codes of the kits used and experimental instruments have been added in the part of Materials and Methods

Point 5: In fig 2 the significance between the various groups is not very clear, please describe better.

Response 5: This part has been written again. And the method to describe the significance has been altered by “Compare to NC group, the fasting serum glucose is significantly lower in MIT group……” This change also adapted the rest of the text.

Point 6: Figure 3A does not correspond to the relative histograms. Furthermore, a higher resolution WB figure should be inserted.

Response 6: In the early stage of the experiment, I took the order of NC, NT, MIIT, HIT, MIT+T, HIT+T to conduct the western blot. But when analysing the statistics, I found the order of NC, NT, MIT, MIT+T, HIT, HIT+T would be more intuitive and clear. So I chose the latter order to present the results. The bands figure is a complete picture, adjusting the order of groups will cause the picture to be cut, so I did not do the processing to keep the original picture.

The higher resolution of WB figure has been inserted, as well as other figures in the article.

Point 7: It would have been interesting to include another endocrine disruptor, what do the authors think about it?

Response 7: This is indeed a great idea to imposing another endocrine disruptor or inhibitor targeting a particular pathway. It will be helpful to elucidate the underlying mechanism more deeply. We will consider that in our future research.

Point 8: The discussion is too dispersive and the pathway in relation to physical activity and glucose metabolism should be better clarified, in this regard the authors could refer to this work (Treatment with sera from Water Polo athletes activates AMPKα and ACC proteins In HepG2 hepatoma cell line).

Response 8: The part of discussion has been reorganized and I removed some redundant sentences. In addition to that, I described the pathway in relation to exercise and glucose metabolism more clearly and cited several important references (including the article that you mentioned which is really helpful). Under the work, the discussion is more clear than before.  

Reviewer 2 Report

Wang et al. investigated the role of the environmental toxin TCDD on glucose metabolism in SD rats, and they determined whether different intensities of exercise can ameliorate the effects of TCDD exposure.  They show that neither moderate- nor high-intensity exercise affect insulin resistance induced by TCDD.  Despite the fact that this article is largely reporting negative data, it is important to understand that, while exercise is a powerful way to treat a number of disorders, it is not necessarily a panacea.  It is also important to draw attention to the fact that TCDD, and, presumably, other environmental toxins, contribute to diseases of metabolism, which represent a significant challenge to human health in the modern world.    

Major comments:

To clarify the exact timing of when the exercise intervention began and when the TCDD was injected, the authors should include a diagram with arrows approximately corresponding to the experimental set up.  In the current text, it is unclear whether the rats began the exercise before, after, or at the same time as the IP injections of TCDD.  This is important for the interpretation of the results, because if the exercise started before the IP injections, then the data means that exercise cannot protect from TCDD exposure.  If the exercise started after the TCDD injection, then it means that exercise cannot reverse the effects of the pollutant.

In comparing the groups in Figure 1, it would be helpful to have more specific information about which groups are being compared.  Listing the p values in the text does not clarify any specific significant differences between any given experimental group compared to the relevant control.  It would also be nice to show the exact p values in situations where the groups were nearly significantly different:  Trends can be informative.

There are multiple typographical or grammatical errors that the authors should correct, particularly in the abstract and figures.

Minor comments:

It is perhaps worth noting that the rats in this study were fed on standard chow.  Given that modern human diets (high sugar/high fat) are more obesogenic, it is an interesting question as to whether the TCDD treatment would have been more detrimental to rats on a “Western diet”.  In this scenario, it may be that exercise would be significantly protective.

Line 94: NC should be “non-exercise”

In the legend of Figure 1, the white circle is labelled “TC”, which should be “NT”.

Line 213-214:  “Figure 3” should be labelled as “Figure 4”

Figure 4A should include the molecular weight markers.

Line 246:  “mainly lied” should be changed to “mainly depended” or similar phrase

Line 264-265:  “which reflecting . . . enhanced.” should be, for example, “reflecting that the function of pancreatic β cells was enhanced.”

Line 309:  “The mediator act of. . . .” should be something like “The action of the IRS/PI3K/Akt signaling. . . .”

Line 319:  “thus be used. . . .” should be “thus it can be used as a marker. . . .”

Author Response

Response to Reviewer 2 Comments

Point 1: To clarify the exact timing of when the exercise intervention began and when the TCDD was injected, the authors should include a diagram with arrows approximately corresponding to the experimental set up. In the current text, it is unclear whether the rats began the exercise before, after, or at the same time as the IP injections of TCDD. This is important for the interpretation of the results, because if the exercise started before the IP injections, then the data means that exercise cannot protect from TCDD exposure. If the exercise started after the TCDD injection, then it means that exercise cannot reverse the effects of the pollutant.

Response 1: The diagram with arrows has been made and inserted in the part of experimental protocol as “Figure 1”. It clarified the exact time of TCDD exposure and exercise. In the discussion part, this point also has been illuminated more clearly combined with the results.

Point 2: In comparing the groups in Figure 1, it would be helpful to have more specific information about which groups are being compared. Listing the p values in the text does not clarify any specific significant differences between any given experimental group compared to the relevant control. It would also be nice to show the exact p values in situations where the groups were nearly significantly different: Trends can be informative.

Response 2: The figure of body weight has been made again to look more clear and aesthetic. The significant differences of body weight between groups in 8th week have been labeled by ** in the figure. However, the significant differences in any week of every group by repeated measurement ANOVA are too much to clearly label. The body weight of any two week in one group almost has significant difference. I tried to labeled all the difference by ** in the figure, but it was difficult to recognize and the figure got in chaos. So I removed them and just left the significant differences between groups in 8th week.

As for the exact p values, all the results that I analyzed in this part are P=0.000, thus I wrote P<0.001 for precise. Other p values with no statistical significance are much larger than 0.05, so I didn’t make change on that. Please pardon me.

Point 3: It is perhaps worth noting that the rats in this study were fed on standard chow. Given that modern human diets (high sugar/high fat) are more obesogenic, it is an interesting question as to whether the TCDD treatment would have been more detrimental to rats on a “Western diet”. In this scenario, it may be that exercise would be significantly protective.

Response 3: This is indeed a great insight to change the rats’ food to explore the metabolic effect of TCDD and exercise on obese rats. The outcomes may be different with the present study. Obesity can leave the body in a state of chronic inflammation over time, TCDD exposing will possibly amplify the state and exercise may be alleviate the toxicity effectively for its obvious anti-inflammatory effect on obese rats. This result will also have more implications for humans. We will consider that in our future research.

Point 4: Line 94: NC should be “non-exercise”; In the legend of Figure 1, the white circle is labelled “TC”, which should be “NT”; Line 213-214: “Figure 3” should be labelled as “Figure 4”; Figure 4A should include the molecular weight markers.; Line 246: “mainly lied” should be changed to “mainly depended” or similar phrase; Line 264-265: “which reflecting . . . enhanced.” should be, for example, “reflecting that the function of pancreatic β cells was enhanced.”; Line 309: “The mediator act of. . . .” should be something like “The action of the IRS/PI3K/Akt signaling. . . .”; Line 319: “thus be used. . . .” should be “thus it can be used as a marker. . . .”

Response 4: All the multiple typographical or grammatical errors as well as the mistakes above have been corrected in the revision.

Reviewer 3 Report

Dear Authors

Congratulation for a very interesting topic, we make you some recommendation that we hope will contribute to improve the clarification of the article and wish you success for the future publication.  

In the end of your introduction please refer in what your study is innovative.

Line 72 you write: inappropriate intensity exercise could even lead to blood glucose elevated, please specify, because this increase is transitory. Future lecturers must not believe that intense exercise could promote a chronic increase of blood glucose.   

Line 99 you write: “they were divided into 6 groups according to their body weight, with no difference among groups” is it enough to be sure that the groups were equivalents, can you discuss this better to elucidate the future lecturers?

3.2. Serum biochemical parameters in rats “Moderate intensity exercise significantly decreased fasting serum glucose”  May be it is better to write compared to NC the fasting serum glucose is significantly inferior in MI group, because in the case of the present study we do not compare the same subjects in two moments. Please adapt the rest of the text!

Author Response

Response to Reviewer 3 Comments

Point 1: In the end of your introduction please refer in what your study is innovative.

Response 1: The innovative point of our research has been added in the end of introduction.

Point 2: inappropriate intensity exercise could even lead to blood glucose elevated, please specify, because this increase is transitory. Future lecturers must not believe that intense exercise could promote a chronic increase of blood glucose.

Response 2: This point and cited reference have all been replaced by a new paper published in “cell metabolism” this year-Excessive exercise training causes mitochondrial functional impairment and decreases glucose tolerance in healthy volunteers. It specifies the reason of glucose tolerance decreases caused by too high intensity may be the mitochondrial functional impairment.

Point 3: “they were divided into 6 groups according to their body weight, with no difference among groups” is it enough to be sure that the groups were equivalents, can you discuss this better to elucidate the future lecturers?

Response 3: In animal experiments, body weight is often the basis for conducting random groupings. The SD rats purchased in our study are all germ-free and healthy, with the same age and gender, thus the body weight is the most effective indicator for randomly assigned rats to achieve homogenization.

Point 4: Serum biochemical parameters in rats “Moderate intensity exercise significantly decreased fasting serum glucose”  May be it is better to write compared to NC the fasting serum glucose is significantly inferior in MI group, because in the case of the present study we do not compare the same subjects in two moments. Please adapt the rest of the text!

Response 4: The part of “result” has been written again. And the method to describe the significance has been altered by “Compare to NC group, the fasting serum glucose is significantly lower in MIT group……” This change also adapted the rest of the text.

Round 2

Reviewer 1 Report

the authors reported all required changes, the manuscript is ok for publication.